# POLARNET: LEARNING TO OPTIMIZE POLAR KEYPOINTS FOR KEYPOINT BASED OBJECT DETECTION

**Xiongwei Wu**[1]**, Doyen Sahoo**[2]**, Steven C.H. Hoi**[1,2]
[1]Singapore Management University
[2]Salesforce Research Asia
`xwwu@smu.edu.sg`
`{dsahoo, shoi}@salesforce.com`

## ABSTRACT

A variety of anchor-free object detectors have been actively proposed as possible alternatives to the mainstream anchor-based detectors that often rely on complicated design of anchor boxes. Despite achieving promising performance on par with anchor-based detectors, the existing anchor-free detectors such as FCOS or CenterNet predict objects based on standard Cartesian coordinates, which often yield poor quality keypoints. Further, the feature representation is also scale-sensitive. In this paper, we propose a new anchor-free keypoint based detector "PolarNet", where keypoints are represented as a set of Polar coordinates instead of Cartesian coordinates. The "PolarNet" detector learns offsets pointing to the corners of objects in order to learn high quality keypoints. Additionally, PolarNet uses features of corner points to localize objects, making the localization scale-insensitive. Finally in our experiments, we show that PolarNet, an anchor-free detector, outperforms the existing anchor-free detectors, and it is able to achieve highly competitive result on COCO test-dev benchmark ($47.8\%$ and $50.3\%$ AP under the single-model single-scale and multi-scale testing) which is on par with the state-of-the-art two-stage anchor-based object detectors. The code and the models are available at `https://github.com/XiongweiWu/PolarNetV1`

## 1 INTRODUCTION

Deep learning based object detection techniques have achieved remarkable success in many real-world applications (Krizhevsky et al., 2012; He et al., 2016; Goodfellow et al., 2016). The mainstream state-of-the-art detectors are often based on the anchor-based detection methods (Ren et al., 2015; Girshick, 2015; Lin et al., 2017b), which heavily rely on the design and selection of appropriate *anchor boxes*, namely a set of predefined bounding boxes of a certain height and width, to capture various scales and aspect ratios of different object classes for detection. Unlike the anchor-based detectors, the anchor-free detectors have emerged recently as a promising direction for object detection that eliminates the need of manually designing anchor boxes (Zhu et al., 2019; Tian et al., 2019; Law & Deng, 2018; Duan et al., 2019).

In literature, a variety of anchor-free object detectors have been proposed based on different object modeling strategies. Figure 1 (a)-(e) gives examples comparing five popular anchor-free detectors from the perspective of object modeling. For example, CornerNet (Law & Deng, 2018) was proposed for detecting objects using a pair of corner points. Instead of using two corners, CenterNet (Zhou et al., 2019a) proposed modeling an object as one center point of its bounding box. Besides these, there are also a number of other anchor-free detectors that extend these ideas of Corner-based or Centerness-based or various other keypoint design strategies to improve the detection performance. FSAF (Zhu et al., 2019) and FCOS (Tian et al., 2019) predict objects by learning the offsets to the boundary from sampled keypoints. FCOS (Tian et al., 2019) uses many keypoints by treating every pixel as a keypoint, while FSAF (Zhu et al., 2019) samples a set of multiple keypoints from the center region to eliminate points near the boundary.

Among keypoint based object detection, two different strategies are commonly adopted. One is keypoint position based (determining the bounding box by the position of the keypoints) such

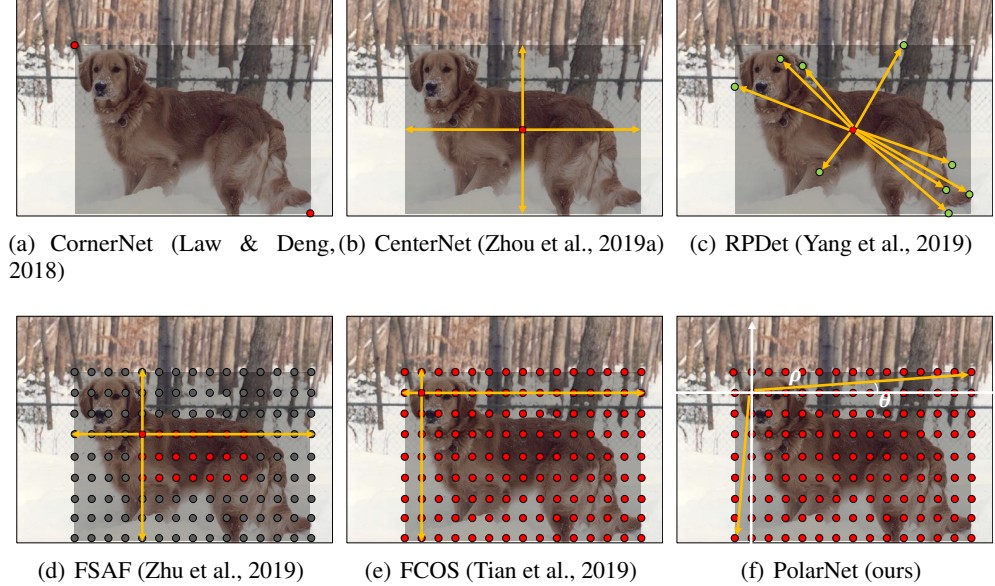

(a) CornerNet (Law & Deng, 2018)    (b) CenterNet (Zhou et al., 2019a)    (c) RPDet (Yang et al., 2019)

(d) FSAF (Zhu et al., 2019)    (e) FCOS (Tian et al., 2019)    (f) PolarNet (ours)

Figure 1: Comparison of different anchor-free object detection methods. Red dots denote positive keypoints and grey dots denote negative keypoints.

CornerNet (Law & Deng, 2018) and CenterNet (Duan et al., 2019). The other is keypoint offsets based (determining the bounding box by learning offsets from keypoints) such as FSAF (Zhu et al., 2019), CenterNet (Zhou et al., 2019a) and FCOS (Tian et al., 2019). Compared to the keypoint position based methods, keypoint offset based methods are able to better capture contextual information about the objects, and are thus more commonly used to design detectors.

However, existing object modeling strategies for keypoint offset based methods may be sub-optimal. Most existing anchor-free detectors such as FCOS (Tian et al., 2019) are based on Cartesian coordinates, which learn offsets to the boundary of objects. However, this kind of design yields a lot of poor quality keypoints. These points are near the boundary with extremely large variance of offsets (See Figure 1(e)). Besides, the prediction heads are also based on the scale-sensitive features, which further increases the optimization difficulties. Our goal is to have an anchor-free detector, which can avoid poor quality keypoints, and is able to simultaneously learn scale-insensitive feature for object prediction. To achieve this goal, in this paper, we propose a new keypoint based object detector named "PolarNet", which learns keypoints based on polar coordinates. Figure 1 (f) illustrates the idea of the proposed PolarNet compared to the other anchor-free detectors. The set of keypoints is represented by polar coordinates to avoid large variance of offsets. And the features of corner points in PolarNet are also used to localize objects, which is scale-invariant.

The key contributions of this work are:

- We introduce a unified view of keypoint based object detection for understanding popular anchor-free object detectors, in which many popular anchor-free object detectors can be viewed as a special case of keypoint based detectors with different object modeling strategies;

- We propose a new anchor-free object detector named "PolarNet" which presents keypoints based on polar coordinates which enables learning of better quality keypoints by reducing the variance of learned offsets compared to the existing approaches.

- We conduct experiments to evaluate the performance of our PolarNet detector on the COCO benchmark, in which our results show that PolarNet outperforms all the existing anchor-free detectors, and is able to achieve highly competitive results better or on par with the state-of-the-art two-stage anchor-based detectors on COCO test-dev (47.8% and 50.3% AP with DCNv2-ResNeXt-101 backbone on COCO test-dev under single-model single-scale and multi-scale settings).

The rest of this paper is organized as follows. Section 2 reviews two major categories of related work in deep-learning based object detection: popular anchor-based detectors and recent anchor-free detectors. Section 3 presents the proposed PolarNet detector in detail. Section 4 discusses our experimental results and analysis, and Section 5 concludes this paper.

## 2 RELATED WORK

In this section, we briefly review two major groups of related work in the literature of deep-learning based object detection approaches: the mainstream family of anchor-based object detection methods and the emerging family of anchor-free object detection methods. We refer readers to more extensive surveys of deep-learning based object detection studies in (Liu et al., 2020; Wu et al., 2020).

### 2.1 ANCHOR-BASED OBJECT DETECTION

The methods in this group represent the mainstream detectors widely used in many real-world applications. They can be broadly categorized into two groups: two-stage detectors (Ren et al., 2015; Girshick et al., 2014; Girshick, 2015) and one-stage detectors (Liu et al., 2016; Zhang et al., 2018; Lin et al., 2017b). Two-stage detectors consist of two stages: (i) region proposal generation, and (ii) region proposal classification and regression. For example, one of most popular two-stage detectors is Faster R-CNN (Ren et al., 2015) that uses a Region Proposal Network (RPN) to generate regions of interest in the first stage and then sends the region proposals down the pipeline for object classification and bounding-box regression. Faster R-CNN has resulted in many various extensions and improvements in literature (Lin et al., 2017a; Dai et al., 2017; 2016). Single-stage detectors, also known as single-shot detectors, do not need the proposal generation and simply treat object detection as a simple classification and regression problem by taking an input image and directly learning the class probabilities and bounding box coordinates of objects via convolutional networks. Popular single-stage detectors include YOLO (Redmon et al., 2016), SSD (Liu et al., 2016), and RetinaNet (Lin et al., 2017b). Typically, single-stage detectors are less accurate than two-stage detectors, but they are much faster and thus more amenable to real-time inference needs. In general, anchor-based detectors suffer from some critical limitations, including requiring heuristic design of anchors, poor alignment of anchors with ground truth objects, and incurring a large number of false positives when anchors are not carefully designed.

### 2.2 ANCHOR-FREE OBJECT DETECTION

Here, we view most anchor-free detectors from as keypoint based detectors. We can categorize the existing anchor-free object detection methods into two groups according to different bounding box prediction strategies: 1) Keypoint Position and 2) Keypoint Offsets based detectors. We review some representative works in each group below.

**Keypoint Position:** This kind of anchor-free detectors predict bounding box via keypoint positions, such as corners. A representative detector is CornerNet (Law & Deng, 2018). CornerNet models each object by a pair of corner keypoints, which eliminates the need of anchor boxes and is perhaps the first anchor-free detector that achieved the state-of-the-art single-stage object detection accuracy. There have been some extensions of CornerNet to improve its efficiency towards real-time applications such as CornerNet-Lite (Law et al., 2019). Based on CornerNet, CenterNet (Duan et al., 2019), models an object by a triplet of keypoints, including one center and two corner keypoints. ExtremeNet (Zhou et al., 2019b) models an object by a set of five keypoints, including one center and four extreme points (top-most, leftmost, bottom-most, right-most) of an object based on a standard keypoint estimation network. RPDet (Yang et al., 2019) represents an object by a fixed set of 9 keypoints learned from the center of an object which are refined progressively during the training process. RPNet V2 (Chen et al., 2020) fuses the corner and foreground features in into the original RPNet to detect objects. Their work is in parallel with ours and shares some similarities with our corner learning module. However, our PolarNet is mainly focused on the design of the polar representation to handle scale issues based on vanilla FCOS, and the corner learning module is an auxiliary module to further enhance the feature representation of the detector.

**Keypoint Offsets:** This kind of anchor-free detectors predict bounding box via learning offsets based on keypoints. A representative anchor-free detector is CenterNet (Zhou et al., 2019a) that models an

object by the center point of its bounding box, and it uses keypoint estimation to find center points and regresses to all other object properties, such as size, 3D location, orientation, and even pose. Some approaches such as FSAS (Zhu et al., 2019) and FoveaBox (Kong et al., 2020) sample many keypoints from the center region of an object and learn to predict objects. Some approaches such as FCOS (Tian et al., 2019) and SAPD (Zhu et al., 2020) treat all the pixels within an object as candidate keypoints during training and improve them by a separate post-processing strategy. These keypoint based detectors all learn offsets to the boundary of the objects based on Cartesian coordinates, which yields a lot of poor quality keypoints. Some strategies are adopted to remove these bad points such as center sampling (Kong et al., 2020; Zhu et al., 2019) or centerness score (Tian et al., 2019). Unlike the above keypoint based detectors that learn keypoints based on Cartesian coordinates, our approach learns a set of keypoints based on polar coordinates which makes it easier to learn high quality keypoints.

## 3   POLARNET DETECTOR

### 3.1   POLAR COORDINATES VS CARTESIAN COORDINATES

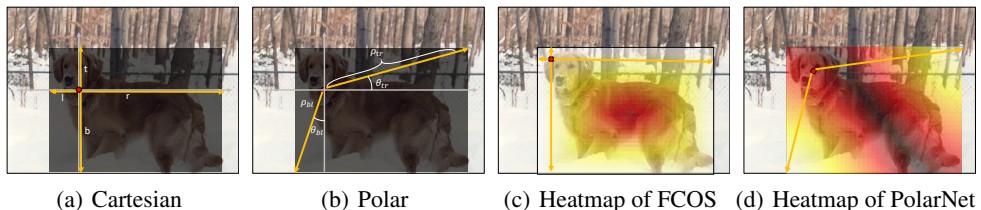

| (a) Cartesian | (b) Polar | (c) Heatmap of FCOS | (d) Heatmap of PolarNet |

Figure 2: (a) and (b) show the comparison of Cartesian coordinates and Polar coordinates. (c) and (d) show the heatmap of learned offsets in training FCOS and PolarNet. In (c), the majority of keypoints learned by FCOS have large scale variance, indicating the necessity to suppress the poor quality keypoints during training. In (d), the scale variance of keypoints in PolarNet is much smaller than FCOS, and thus more high quality keypoints can be learned during training.

Many popular keypoint based detectors represent keypoints based on Cartesian coordinates. This strategy yields a lot of keypoints with poor quality (see Figure 1(e)), making it difficult to optimize without manually designed strategies (such as centerness (Tian et al., 2019) or center sampling (Zhu et al., 2019)). A better approach for keypoints representation is thus required to avoid poor quality keypoints (such as points near the boundary). In order to overcome this challenge, we represent the keypoints with polar coordinates instead of vanilla Cartesian coordinates. We learn the offsets of each keypoints pointing to the corners, which significantly reduce the scale variance of keypoints. Furthermore, the features of corners are also used for scale-insensitive localization.

For each keypoint, we predict offsets based on this point to generate the bounding box. Vanilla keypoint based detectors learn keypoints based on Cartesian coordinates, which learns 4D vectors $(t, l, b, r)$, corresponding to (top, bottom, left, right) pointing to the boundary of the objects (See Figure 2(a)). In PolarNet, for each object, we learn the offsets of keypoints which point to the corners (In this paper we use top left (tl) and bottom right (br) as the target corner pair). And thus for each point $(x, y)$, we predict a pair of 4D vectors: $(\theta_{tl}, \rho_{tl}, \theta_{br}, \rho_{br})$, which point to the corners of the object (See Figure 2(b)). Here $\rho$ and $\theta$ denote the distance and sine value of angle of point $(x, y)$ to corner. We defer the discussion for training the keypoint predictor later.

For keypoint offsets based detectors, learning offsets with large variance yields unsatisfactory results. And thus in many existing methods, poor quality points will be removed (Zhu et al., 2019) or be suppressed (Tian et al., 2019) during training. Here we analyze the scale variance of the learned offsets of the keypoints in FCOS and in PolarNet during training. Assume in FCOS and PolarNet, the learned offsets of keypoint $(x, y)$ are represented as $\mathbf{B_{x,y}}$. To show the scale variance of learned offsets, We plot the heatmaps of scale variance of the learned offsets of FCOS and PolarNet in Figure 2(c) and (d) respectively as:

$$k = \frac{\min\{\mathbf{B_{x,y}}\}}{\max\{\mathbf{B_{x,y}}\}} \tag{1}$$

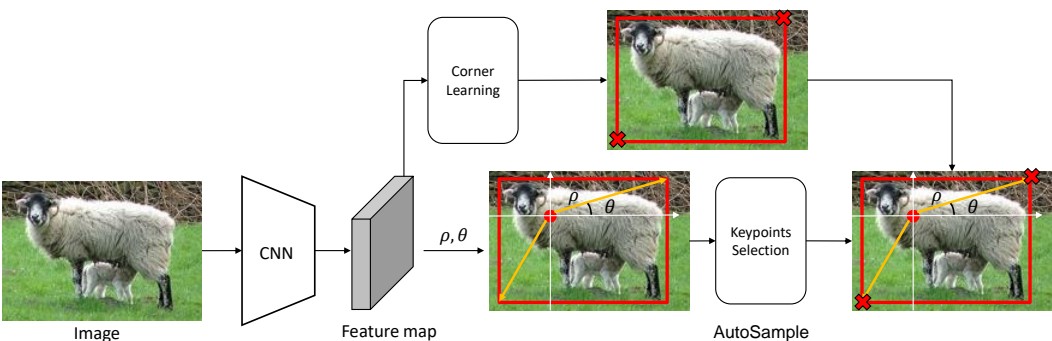

Figure 3: The architecture of our PolarNet detector. Following the CNN backbone, we also apply the Feature Pyramid Network (FPN). The objectness prediction module is a pixel-wise prediction to predict the likelihood of a relevant object in the location of the pixel. Each keypoint is represented by polar coordinates which points to corners. The optimal coordinates are selected for training and the features of corners are also used for training detectors.

where $k$ is the value plotted in heatmaps. In Figure 2(c), nearly all the keypoints within the objects present a large scale variance except the ones within the central region. Specifically when the point is close to the boundary, the variance of learned offsets is very large. In Figure 2(d), the majority of the keypoints in PolarNet learn offsets with much smaller scale variance compared with Figure 2(c), which significantly reduce the optimization difficulties.

## 3.2 OVERVIEW

We now present the proposed keypoint based detection network (PolarNet), a new anchor-free detector that is capable of better representing keypoints and localizing the objects. Figure 3 gives an overview of the architecture of the proposed PolarNet. An input image is passed through a CNN network to produce a set of feature maps. Note that following the CNN backbone, we also apply the Feature Pyramid Network (FPN) to help deal with scale variances. Based on the feature maps, a Fully Convolutional layer is applied to perform the pixel-wise objectness prediction, which predicts the objectness likelihood of a pixel with respect to a particular object category. Then for each keypoint within the objects, we will learn a set of polar coordinates pointing to the corners of the objects. The keypoints selector will be used here to select the optimal coordinates for detector training. And the features of corners will also be used here to supervise the detector learning. Finally, after the selected keypoints are obtained, the keypoints' bounding boxes together with their objectness and likelihood scores are passed through a Non-Maximum Suppression (NMS) to obtain the final detection result.

Next we will discuss several key modules of the proposed framework, including polar coordinates, pixel-wise objectness prediction, overall detector training, inferences, and other details.

## 3.3 PIXEL-WISE OBJECTNESS PREDICTION

The input to this module is a set of feature maps from a CNN backbone, denoted by $F_i \in \mathbb{R}^{H \times W \times C}$ the feature map at layer $i$ of a CNN with a total of $C$ classes. We view any pixel/location $(x, y)$ of the feature map as a potential candidate for keypoints, and the objectness prediction module aims to predict how likely a particular location of the feature map is relevant to some particular class of objects. This idea follows the principle of fully convolutional networks (FCN) for semantic segmentation and has also been used previously in FCOS (Tian et al., 2019).

Specifically, we predict the objectness of a particular location $(x, y)$ by a real vector $(\mathbf{c}_{x,y}, \mathbf{b}_{x,y})$, where $\mathbf{c}_{x,y} \in (0, 1)^C$ denotes a $C$-dimensional vector of object class prediction scores for $(x, y)$.

## 3.4 DETECTOR TRAINING

### 3.4.1 TRAINING LOSS FOR KEYPOINTS CLASSIFICATION

We train the objectness prediction using both classification loss and bounding box regression loss. We define $\mathcal{Q} = (x, y)$ is the set of candidate locations that fall into the ground-truth bounding boxes.

The classification loss is defined as

$$L_{\text{cls}} = \frac{1}{|\mathcal{Q}|} \sum_{(x,y)} L_{\text{focal}}(\mathbf{c}_{x,y}, \mathbf{c}_{x,y}^*)  \tag{2}$$

where $L_{\text{focal}}$ is based on the focal loss (Lin et al., 2017b) and $\mathbf{c}_{x,y}^* \in [0,1]^C$ denotes the ground-truth class labels at location $(x, y)$. For bounding box regression, we only consider a location that falls into any ground-truth box. Unlike vanilla keypoint detectors which yield poor quality keypoints due to large variance of learned offsets, in our model the majority of keypoints have similar distance $\rho$ to the keypoints and thus it is more suitable for training the detectors. We can still further improve the detector algorithms with the keypoints sampling methods, which removes the remaining poor quality keypoints around the two corners. Here we validate a few sampling strategies and select a simple yet effective strategy, AutoSample in our final model. Please refer Appendix .1 for more analysis.

### 3.4.2 TRAINING LOSS FOR CORNER FEATURE LEARNING

In vanilla keypoint detector, the learned features for prediction are mainly based on FPN and are sensitive to scales. Inspired by the scale-invariant property of corner features, we aim to enhance the original feature representations based on the corner features. For each feature map of the feature pyramid, we first learn the corner feature supervised by the corner annotation, and then integrate the learned corner feature into the original feature map to make it robust to scale variance. We follow the similar training strategy and define the corner loss $L_{\text{corner}}$ as CornerNet (Law & Deng, 2018) and we learn detectors based on the new generated feature maps. Please refer Appendix .2 for more details.

### 3.4.3 TRAINING LOSS FOR LOCALIZATION

Here we train the bbox regressors based on the new generated features maps. Suppose the polar coordinates in $(x, y)$ is $\mathbf{B}_{\mathbf{x},\mathbf{y}}$ as $(\theta_{\text{br}}, \rho_{\text{br}}, \theta_{\text{tl}}, \rho_{\text{tl}})$, We can easily obtain the bounding box based on the polar coordinate. And we define the bounding box regression loss as:

$$L_{\text{bbox}} = \frac{1}{|\mathcal{Q}|} \sum_{(x,y) \in \mathcal{Q}} L_{\text{GIoU}}(\mathbf{B}_{x,y}, \mathbf{B}_{x,y}^*)  \tag{3}$$

where $L_{\text{GIoU}}$ is based on the GIoU loss (Rezatofighi et al., 2019). We transform the learned offsets into bounding box and optimize the GIoU loss. In addition to GIoU loss, here we also learn the supervision from corners as:

$$L_{\text{theta}} = \frac{1}{2|\mathcal{Q}|} \sum_{(x,y)} \sum_{p \in (\text{tl,br})} L_{\text{CE}}(\sin \theta_p, \sin \theta_p^*)  \tag{4}$$

where $L_{\text{CE}}$ denotes cross entropy loss. We find obtaining the supervision from $\theta$ is sufficient to train the model and thus we do not learn supervision from $\rho$. Finally, we define localization loss as:

$$L_{\text{loc}} = L_{\text{bbox}} + L_{\text{theta}}  \tag{5}$$

### 3.4.4 OVERALL TRAINING LOSS AND ADAPTIVE KEYPOINT TRAINING

Combining the above training losses for both objectness prediction and polar coordinates modules, we can define the overall training loss function as follows:

$$L_{\text{overall}} = L_{\text{cls}} + L_{\text{loc}} + L_{\text{corner}}  \tag{6}$$

### 3.5 INFERENCE

During the inference stage, PolarNet takes an image as an input and passes it through the CNN network followed by the FPN module to obtain the feature maps. Based on the resulting feature maps, we then predict the objectness scores $\mathbf{c}_{x,y}$ and bounding box offsets $\mathbf{b}_{x,y}$ of each point/location $(x, y)$ on the feature map. After that, we select top-k points/locations with the highest class objectness scores as the candidate points to the keypoint prediction module. Our model shares the same computation cost as vanilla FCOS except the corner learning module. Finally, we pass the predicted boxes to a Non-Maximum Suppression (NMS) to obtain the final detection result.

## 4 EXPERIMENTS

### 4.1 EXPERIMENTAL DATASET AND SETUP

We conducted experiments on MSCOCO 2017 dataset, which has 80 categories in three splits: train (115k images), val (5k images), and test-dev (20k images). Following common practice, we used the train set to train our model and the val set for ablation studies, and finally report the results on test-dev set for comparison. In our experiments, only bounding box level annotations are used. We consider three types of backbones: ResNet-50 (He et al., 2016), ResNet-101 (He et al., 2016) and ResNeXt-101-DCNv2(Xizhou Zhu & Dai, 2019). For efficiency, ResNet-50 is used in our ablation study.

### 4.2 IMPLEMENTATION DETAILS

Following the recent state-of-the-art object detectors such as FCOS (Tian et al., 2019) and RetinaNet (Lin et al., 2017b), we adopt the ResNet (He et al., 2016) and ResNeXt (Xie et al., 2017) CNN network as our backbone architecture. ResNet and ResNeXt are two fully convolutional networks, which are composed of a sequence of residual modules and were first used for image classification. Residual module first encodes the input feature by a sequence of convolution and normalization layers, and then aggregates the generated feature map with the original input features. In order to predict objects with large scale variance, we also apply the Feature Pyramid Network (FPN) (Lin et al., 2017a) in our approach, which combines the shallow layer features with deep layer features by the latent connection. To learn a scale-robust detector, each level of FPN is responsible for a certain scale of objects, making it very suitable for object detection. Specifically, we use 5 FPN levels to make prediction, with stride 8, 16, 32, 64 and 128 compared with the original image, and each of the level is responsible for a certain scale of the objects: (0, 64], (64, 128], (128, 256], (256, 512] and (512, INF]. We adopt ResNet-50 (He et al., 2016), ResNet-101 (He et al., 2016) and DCN2-ResNeXt-101-64x4 (Xizhou Zhu & Dai, 2019) as our backbone architecture in our experiments.

We train the model from weights pre-trained on ImageNet classification task and other parameters are initialized by the same methods as RetinaNet (Lin et al., 2017b). The model is trained with SGD optimization methods with 180k iterations with 16 images per mini-batch. The initial learning rate is set to 1e-2 and is reduced 10 times at 120k and 160k iterations. We re-scale the input images into 800x1333 pixels before training. We use the same data augmentation strategy presented in (Tian et al., 2019) when training the model, and for each image, the top-100 predictions are produced.

### 4.3 EXPERIMENTAL RESULTS

Table 1 shows the results on the COCO val set by comparing our method with other popular anchor-free detectors mostly with ResNet-101 (for CornerNet and CenterNet we add results on Hourglass-104 since they are designed based on Hourglass backbones). For comparison purposes, we also include two anchor-based detectors, including the popular RetinaNet (Lin et al., 2017b), TSD (Song et al., 2020) and the state-of-the-art ATSS (Zhang et al., 2020b). We also include the new proposed detector DETR, which is based on the transformer (Vaswani et al., 2017) and shows highly competitive results.

From the results, we can see that all anchor-free/keypoint-based methods outperform RetinaNet which uses predefined anchors and IoU matching methods. This confirms the advantage of keypoint-based detection methods over heuristic anchor-based designs. However, the existing anchor-free detectors are worse than ATSS, which is a recent state-of-the-art anchor-based method by borrowing and adapting some advanced strategies from anchor-free methods. By examining the results of our PolarNet, we found that it outperforms all the existing keypoint-based detectors. Compared with Keypoint Position methods such as CornerNet, our model is able to capture more contextual information of the objects. Compared with other Keypoint Offsets methods such as FCOS, our method represents keypoints by polar coordinates which eliminate poor keypoints and learn box via corner supervision. This merit significantly boosts the performance of our method. Finally, our method is even also better than the state-of-the-art anchor-based ATSS, but eliminates the need of manually designed anchors. DETR is the new proposed detection algorithm based on transformer which automatically learns the attention between each position of the image, and it predicts objects via a sequence of query position embedding. The transformer-based detection algorithm shows the great power to handle vision problems, but our method is still better than the DETR.

| Object Detectors | Anchor-free | Backbone | AP | $AP_{50}$ | $AP_{75}$ | $AP_S$ | $AP_M$ | $AP_L$ |
|---|---|---|---|---|---|---|---|---|
| RetinaNet (Lin et al., 2017b) | Anchor-based | R-101 | 39.1 | 59.1 | 42.3 | 21.8 | 42.7 | 50.2 |
| TSD (Song et al., 2020) | Anchor-based | R-101 | 42.3 | 63.1 | 45.9 | 25.1 | 46.3 | 56.5 |
| ATSS (Zhang et al., 2020b) | Anchor-based | R-101 | 43.5 | 62.1 | 47.4 | 26.1 | 47.0 | 53.6 |
| CornerNet (Law & Deng, 2018) | Keypoint Position | R-101 | 30.2 | 44.1 | 32.0 | 13.3 | 33.3 | 42.7 |
| CornerNet (Law & Deng, 2018) | Keypoint Position | HG-104 | 40.5 | 56.5 | 43.1 | 19.4 | 42.7 | 53.9 |
| RPDet (Yang et al., 2019) | Keypoint Position | R-101 | 41.0 | 62.9 | 44.3 | 23.6 | 44.1 | 51.7 |
| CenterNet (Zhou et al., 2019a) | Keypoint Offsets | R-101 | 34.6 | 53.0 | 36.9 | - | - | - |
| CenterNet (Zhou et al., 2019a) | Keypoint Offsets | HG-104 | 42.1 | 61.1 | 45.9 | 24.1 | 45.5 | 52.8 |
| FSAF (Zhu et al., 2019) | Keypoint Offsets | R-101 | 40.9 | 61.5 | 44.0 | 24.0 | 44.2 | 51.3 |
| FoveaBox (Kong et al., 2020) | Keypoint Offsets | R-101 | 40.6 | 60.1 | 43.5 | 23.3 | 45.2 | 54.5 |
| FCOS (Tian et al., 2019) | Keypoint Offsets | R-101 | 41.5 | 60.7 | 45.0 | 24.4 | 44.8 | 51.6 |
| DETR (Carion et al., 2020) | - | R-101 | 43.5 | 63.8 | 46.4 | 21.9 | 48.0 | 61.8 |
| PolarNet (ours) | Keypoint Offsets | R-101 | **43.9** | 62.6 | 47.5 | 26.7 | 47.5 | 56.8 |

Table 1: Performance evaluation of popular keypoint based detectors and two anchor-baesd detectors. The models are trained on MSCOCO train with 115k images, and they are validated on MSCOCO val set with 5k images. "R-101" denotes ResNet-101 backbone and "HG-104" denotes Hourglass-104 backbone. For the backbone architecture, ResNet-101 (R-101) is pre-trained on ImageNet, while Hourglass network (HR-104) is trained from scratch.

## 4.4 ABLATION STUDY

This ablation study aims to examine the effectiveness of polar coordinates and corner learning in our PolarNet. Table 2 shows the results of our ablation study.

We first examine the effectiveness of polar coordinates. We set vanilla FCOS as baseline and compare the baseline with FCOS with polar coordinates (we denote this FCOS variant as FCOS-Polar). Notably, we only change the encoding method of vanilla FCOS with Polar coordinates, without any additional computation cost. In vanilla FCOS, the gradient of the localization error of each keypoint is suppressed by its corresponding centerness score, to reduce the impact of large-scale variance. And thus we first train the vanilla FCOS and FCOS-Polar without the centerness. From the results in Table 2, FCOS cannot even converge if trained without the centerness, while our FCOS-Polar can still achieves a highly competitive results (37.4%), which is even comparable with the FCOS trained with centerness score (37.6%). This experiment indicates the effectiveness of the polar coordinates to handle scale issue. Then we train FCOS-Polar with centerness score, and this variant can still outperform vanilla FCOS (38.0% vs 37.6%). Finally, we train our FCOS-Polar based on the AutoSample strategy, and this variant achieves 38.9% AP on COCO metric, which even outperforms the best FCOS variant (38.5%) without any additional feature enhancement.

We then explore the impact of corner features. We integrate corner features with vanilla FCOS (denoted as FCOS-Corner), and this variant achieves 38.4% on COCO metric, which significantly improves the accuracy (38.4% vs 37.6%), indicating the effectiveness of corner learning module. Finally, we learn our PolarNet and it achieves 39.6% AP on COCO metric, which is significantly better than the best FCOS variant and it is also a highly competitive result among all anchor-free detection methods.

| Methods | AP | $AP_{50}$ | $AP_{75}$ | $AP_S$ | $AP_M$ | $AP_L$ |
|---|---|---|---|---|---|---|
| FCOS w/o Centerness (Tian et al., 2019) | *fail to converge* | - | - | - | - | - |
| FCOS w/ Centerness (Tian et al., 2019) | 37.6 | 56.1 | 40.8 | 22.1 | 41.4 | 48.8 |
| FCOS-Polar w/o Centerness | 37.4 | 56.3 | 40.0 | 22.0 | 41.2 | 48.0 |
| FCOS-Polar w/ Centerness | 38.0 | 56.7 | 40.8 | 22.3 | 41.7 | 49.2 |
| FCOS-Polar w/ AutoSample | 38.9 | 57.5 | 41.8 | 22.4 | 43.0 | 50.5 |
| FCOS-Corner w/ Centerness | 38.4 | 56.8 | 41.1 | 22.9 | 42.4 | 49.5 |
| FCOS w/ imprv (Tian et al., 2019) | 38.5 | 57.4 | 41.4 | 22.3 | 42.5 | 49.8 |
| PolarNet | **39.6** | 57.9 | 42.9 | 23.2 | 43.5 | 51.8 |

Table 2: Ablation study on different components in PolarNet. Models are trained on COCO train2017 and tested on COCO val2017 with ResNet-50.

### 4.5 COMPARISON WITH STATE-OF-THE-ART DETECTORS

We compare PolarNet with other state-of-the-art detectors on COCO test-dev set. Unlike the previous experiments, we train our models on backbone ResNeXt-101-DCNv2. Table 3 shows the results on COCO test-dev under the single-model settings, and the inference time is based on Titan-Xp. PolarNet outperforms all single-stage detectors by a substantial margin (except ATSS), also outperform a variety of two-stage/multi-stage detectors, and it achieves the best results among all the keypoints based detectors. This promising result validates our hypothesis of learning keypoints based on polar coordinates with corner supervision is able to better predict the objects. Specifically, compared with other center-based methods such as CenterNet or FCOS, our method not only extracts features from the central region, but also encodes features from other informative areas based on the shape of the objects. Besides, our corner supervision is scale-invariant and thus shows better results in localization. Finally, compared with the SOTA anchor-based ATSS, our method eliminates the need of using anchors and thus avoids heuristic anchor design.

| Method | Backbone | FPS | AP | $AP_{50}$ | $AP_{75}$ | $AP_S$ | $AP_M$ | $AP_L$ |
|---|---|---|---|---|---|---|---|---|
| **Anchor-based** | | | | | | | | |
| **Multi-stage** | | | | | | | | |
| FRCN-FPN (Lin et al., 2017a) | R-101 | 9.9 | 36.2 | 59.1 | 39.0 | 18.2 | 39.0 | 48.2 |
| Fitness-NMS (Tychsen-Smith & Petersson, 2018) | R-101 | 5.0 | 41.8 | 60.9 | 44.9 | 21.5 | 45.0 | 57.5 |
| Cascade R-CNN(Cai & Vasconcelos, 2018) | R-101 | 8.0 | 42.8 | 62.1 | 46.3 | 23.7 | 45.5 | 55.2 |
| Libra R-CNN (Pang et al., 2019) | X-101-64x4d | 5.6 | 43.0 | 64.0 | 47.0 | 25.3 | 45.6 | 54.6 |
| FreeAnchor (Zhang et al., 2019) | X-101-32x8d | 5.4 | 44.8 | 64.3 | 48.4 | 27.0 | 47.9 | 56.0 |
| TridentNet (Li et al., 2019) | R-101-DCN | 1.3 | 46.8 | 67.6 | 51.5 | 28.0 | 51.2 | 60.5 |
| Dynamic R-CNN (Zhang et al., 2020a) | R-101 | - | 42.0 | 60.7 | 45.9 | 22.7 | 44.3 | 54.3 |
| Cascade R-CNN w/ SABL (Wang et al., 2020) | R-101 | 8.8 | 43.3 | 60.9 | 46.2 | 23.8 | 46.5 | 55.7 |
| TSD++ (Song et al., 2020) | X-101-64x4d-DCN | - | 49.2 | 70.1 | 53.8 | 33.2 | 53.1 | 63.7 |
| **Single-stage** | | | | | | | | |
| RetinaNet (Lin et al., 2017b) | R-101 | 8.0 | 39.1 | 59.1 | 42.3 | 21.8 | 42.7 | 50.2 |
| AB+FSAF (Zhu et al., 2019) | R-101 | 7.1 | 40.9 | 61.5 | 44.0 | 24.0 | 44.2 | 51.3 |
| AB+FSAF (Zhu et al., 2019) | X-101-64x4d | 4.2 | 42.9 | 63.8 | 46.3 | 26.6 | 46.2 | 52.7 |
| M2Det (Zhao et al., 2019) | VGG-16 | 11.8 | 41.0 | 59.7 | 45.0 | 22.1 | 46.5 | 53.8 |
| ATSS (Zhang et al., 2020b) | X-101-64x4d-DCN | 7.1 | 47.7 | 66.5 | 51.9 | 29.7 | 50.8 | 59.4 |
| **Anchor-free** | | | | | | | | |
| GA-RetinaNet (Wang et al., 2019) | R-50 | 10.8 | 37.1 | 56.9 | 40.0 | 20.1 | 40.1 | 48.0 |
| GA-FRCN (Wang et al., 2019) | R-50 | 9.4 | 39.8 | 59.2 | 43.5 | 21.8 | 42.6 | 50.7 |
| ExtremeNet (Zhou et al., 2019b) | HG-104 | 2.8 | 40.2 | 55.5 | 43.2 | 20.4 | 43.2 | 53.1 |
| CornerNet (Law & Deng, 2018) | HG-104 | 3.1 | 40.5 | 56.5 | 43.1 | 19.4 | 42.7 | 53.9 |
| FCOS (Tian et al., 2019) | R-101 | 9.3 | 41.5 | 60.7 | 45.0 | 24.4 | 44.8 | 51.6 |
| CenterNet (Zhou et al., 2019a) | HG-104 | 7.8 | 42.1 | 61.1 | 45.9 | 24.1 | 45.5 | 52.8 |
| CenterNet (Duan et al., 2019) | HG-104 | 3.3 | 44.9 | 62.4 | 48.1 | 25.6 | 47.4 | 57.4 |
| RPDet (Yang et al., 2019) | R-101-DCN | 8.0 | 45.0 | 66.1 | 49.0 | 26.6 | 48.6 | 57.5 |
| FCOS w/ imprv(Tian et al., 2019) | X-101-64x4d-DCN | 4.7 | 46.5 | 65.7 | 50.6 | 28.9 | 49.2 | 58.1 |
| FoveaBox (Kong et al., 2020) | R-101 | 11.2 | 40.6 | 60.1 | 43.5 | 23.3 | 45.2 | 54.5 |
| FoveaBox (Kong et al., 2020) | X-101 | - | 42.1 | 61.9 | 45.2 | 24.9 | 46.8 | 55.6 |
| SAPD (Zhu et al., 2020) | R-101 | 11.2 | 43.5 | 63.6 | 46.5 | 24.9 | 46.8 | 54.6 |
| SAPD (Zhu et al., 2020) | X-101-64x4d-DCN | 4.5 | 47.4 | 67.4 | 51.1 | 28.1 | 50.3 | 61.5 |
| CentripetalNet (Dong et al., 2020) | HG-104 | - | 45.8 | 63.0 | 49.3 | 25.0 | 48.2 | 58.7 |
| CentripetalNet++ (Dong et al., 2020) | HG-104 | - | 47.8 | 65.0 | 51.5 | 28.9 | 50.2 | 59.4 |
| CPN (Duan et al., 2020) | HG-104 | 7.3 | 47.0 | 65.0 | 51.0 | 26.5 | 50.2 | 60.7 |
| CPN++ (Duan et al., 2020) | HG-104 | - | 49.2 | 67.4 | 53.7 | 31.0 | 51.9 | 62.4 |
| **PolarNet** (ours) | R-101 | 6.7 | **44.1** | 63.0 | 48.0 | 26.5 | 47.2 | 54.3 |
| **PolarNet** (ours) | X-101-64x4d-DCN | 4.3 | **47.8** | 67.0 | 52.1 | 29.7 | 50.9 | 59.5 |
| **PolarNet++** (ours) | X-101-64x4d-DCN | - | **50.3** | 69.0 | 55.4 | 33.5 | 53.5 | 61.5 |

Table 3: Comparison of PolarNet with other state-of-the-art two-stage or one-stage detectors (*single-model results*). All models were trained on COCO train set and tested on test-dev set. The "++" means the improved inference version of the proposed algorithm, i.e., inference with multi-scale and aspect-ratios, which is widely used in literature.

## 5 CONCLUSION

In this paper, we presented a unified view of popular anchor-free object detectors from the object modeling based detection perspective. We argue that the existing keypoint based detectors often yield a lot of poor quality keypoints based on Cartesian coordinates and the learning of detector is also based on scale-variant features. To overcome the limitation, we proposed PolarNet, a new keypoint based detector by representing keypoints using polar coordinates which avoids poor quality keypoints as well as learning scale-invariant corner features for object localization. Our new detector obtained the highly competitive results on the MS COCO test-dev set under the single-model single-scale and multi-scale settings.

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

APPENDIX

.1 ANALYSIS ON KEYPOINTS SELECTION

From Figure 2, although polar system has significantly reduced the number of poor quality keypoints, we can still further improve the detection algorithms. In this section, we aim to explore the strategy to further reduce the impact of these poor quality keypoints. We have three keypoint selection strategies shown in Figure 4:

- FCOS-Polar w/ dual-head (Figure 4(a)). Two polar heads are learned in FCOS-Polar which point to different corner pairs, and the variance of learned offsets are also learned. During inference, the head with smaller offsets variance will be selected.

- FCOS-Polar w/ CenterSample (Figure 4(b)). Vanilla center sampling strategy adopted in FCOS (Tian et al., 2019), where only the small central regions of the objects will be used which is determined by the FPN stride.

- FCOS-Polar w/ AutoSample (Figure 4(c)). We adopt a shape-aware sampling strategy which samples keypoints with sufficient centerness score (we set threshold as $\gamma$).

From Table 4, all variants show some improvement compared with the baseline, and we found FCOS-Polar w/ AutoSample achieves best results. Compared with the other two variants, autosample is a parameter-free strategy and it is able to learn high quality keypoints based on the shape of the objects. The central region of vanilla center sampling is determined by the FPN stride, which has limited relationship with the shape of different objects, and thus it fails to capture sufficient contexts of the objects. While AutoSample not only reduces the poor quality keypoints but also keep sufficient training samples based on the shape of different objects.

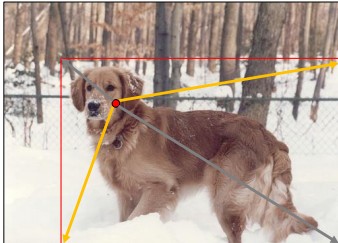 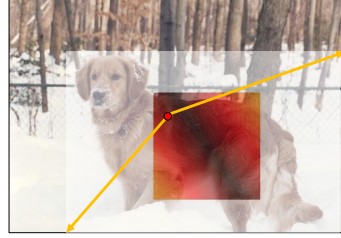 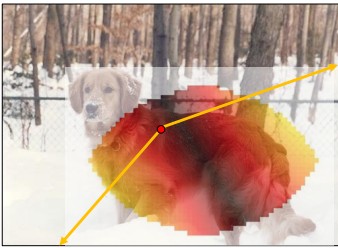

(a) FCOS-Polar w/ dual-head     (b) FCOS-Polar w/ CenterSample     (c) FCOS-Polar w/ AutoSample

Figure 4: (a), (b) and (c) show the comparison of three keypoints selection strategies. The model is trained on COCO train2017 and tested on COCO val2017. All the models are based on ResNet-50 without corner feature enhancement.

We further conduct the ablation study to explore the impact of $\gamma$ in AutoSample. The results are shown in Table 5 and the heatmaps of different $\gamma$ are shown in Figure 5. From the heatmaps, the variants of AutoSample show a good trade-off between rejecting poor quality keypoints and keeping sufficient information compared with vanilla center sampling, and they also obtain significant improvement. Based on the results, we select $\gamma$ as 0.4 in all our experiments.

| Methods | AP | $AP_{50}$ | $AP_{75}$ | $AP_S$ | $AP_M$ | $AP_L$ |
|---|---|---|---|---|---|---|
| FCOS-Polar | 38.0 | 56.7 | 40.8 | 22.3 | 41.7 | 49.2 |
| FCOS-Polar w/ dual-head | 38.3 | 57.0 | 41.4 | 22.4 | 42.1 | 49.4 |
| FCOS-Polar w/ CenterSample | 38.6 | 57.1 | 42.0 | 22.6 | 42.2 | 50.2 |
| FCOS-Polar w/ AutoSample | **38.9** | 57.5 | 41.8 | 22.4 | 43.0 | 50.5 |

Table 4: Ablation study on different keypoints sampling strategy in PolarNet. Models are trained on COCO train2017 and tested on COCO val2017 with ResNet-50.

| Value of $\gamma$ | AP | $AP_{50}$ | $AP_{75}$ | $AP_S$ | $AP_M$ | $AP_L$ |
|---|---|---|---|---|---|---|
| 0.0 | 38.0 | 56.7 | 40.8 | 22.3 | 41.7 | 49.2 |
| 0.2 | 38.2 | 56.9 | 41.1 | 22.4 | 42.0 | 49.4 |
| 0.3 | 38.7 | 57.4 | 41.7 | 21.8 | 42.7 | 50.3 |
| 0.4 | **38.9** | 57.5 | 41.8 | 22.4 | 43.0 | 50.5 |
| 0.5 | 38.3 | 56.4 | 41.1 | 21.6 | 42.4 | 50.2 |
| 0.6 | 37.8 | 55.9 | 40.6 | 22.0 | 41.6 | 50.0 |

Table 5: Ablation study on different values of $\gamma$ in AutoSample. Models are trained on COCO train2017 and tested on COCO val2017 with ResNet-50.

### .2 CORNER FEATURE LEARNING

Corner features are scale-invariant, and thus are particularly suitable to enhance the feature representation of our PolarNet. In addition, the corner features can further enhance the feature representation around the corners where keypoints around are difficult to optimize. We follow the similar training strategy as CornerNet (Law & Deng, 2018) to train the corner features as:

$$L_{\text{corner-cls}} = \frac{-1}{N} \sum_{i=1}^{H} \sum_{j=1}^{W} \begin{cases} (1 - p_{ij})^\alpha \log(p_{ij}) & \text{if } y_{ij} = 1 \\ (1 - y_{ij})^\beta (p_{ij})^\alpha \log(1 - p_{ij}) & \text{otherwise} \end{cases} \tag{7}$$

$$L_{\text{corner-loc}} = \frac{1}{N} \sum_{k=1}^{N} \text{SmoothL1Loss}(\boldsymbol{o}_k, \hat{\boldsymbol{o}}_k) \tag{8}$$

$$L_{\text{corner}} = L_{\text{corner-cls}} + L_{\text{corner-loc}} \tag{9}$$

where $N$, $H$ and $W$ denote the corner number, height and width of the feature map, while $p_{ij}$ and $y_{ij}$ denote the confidence score and binary class label of position $(i, j)$. And $\boldsymbol{o}_k$ and $\hat{\boldsymbol{o}}_k$ denote the learned and target offset of corner $k$ from down sampled feature map to the original image. After the features are learned, we integrate the learned scale-invariant corner features into the original PolarNet features to make it scale-insensitive. In our final model, foreground features are also used to further improve the results.

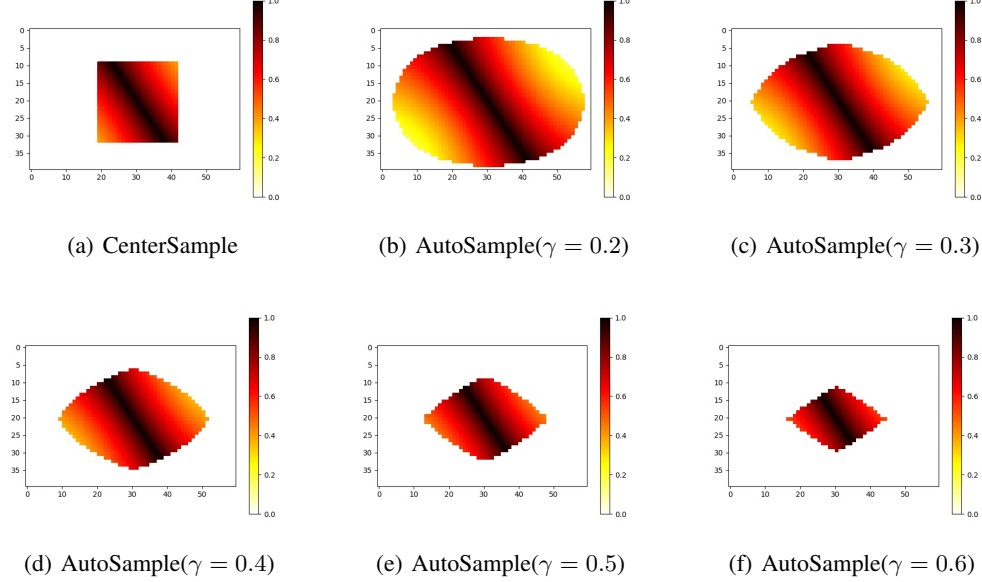

(a) CenterSample     (b) AutoSample($\gamma = 0.2$)     (c) AutoSample($\gamma = 0.3$)

(d) AutoSample($\gamma = 0.4$)     (e) AutoSample($\gamma = 0.5$)     (f) AutoSample($\gamma = 0.6$)

Figure 5: (a) shows the heatmap of CenteSample, and (b)-(f) show the heatmaps of AutoSample based on different $\gamma$.

