# OpenReview forum: "PolarNet: Learning to Optimize Polar Keypoints for Keypoint Based Object Detection"
_ICLR.cc/2021/Conference — ICLR 2021 Poster_

### Official Review · AnonReviewer4 · 2020-10-27
**Official Blind Review #4**

**Rating:** 6
**Confidence:** 4

**Review:**

Summary:

In this paper, the author proposes a new anchor-free keypoint based detector, which learns keypoints based on polar coordinates. It can avoid the large variance of learned offsets compared to the existing anchor-free detectors and make bounding box prediction scale-invariant.

Reasons for score:

Overall, I vote for accepting. My major concern is about the clarity of the paper and some additional analysis (see cons below). Hopefully the authors can address my concern in the rebuttal period.

Pros:

1.This paper proposes a new keypoint based object detector, which represents keypoints based on polar coordinates. It can avoid poor quality keypoints suffered by the existing keypoint based detectors and make regression scale-invariant.

2.Experiments are well thought out and highlight the key advantages of the method over other keypoint based detectors.

Cons:

1.In the Section 3.4.2, why use tangent function for angle regression? For tan, its derivative is $\sec^{2}$. When $|\theta_{br}-\theta_{br}^{*}| \to \frac{\pi}{2}$, the gradient is very large. Will this condition happen? In addition, are there some constraints or operations on $\theta$ to make $\theta \in (0, \frac{\pi}{2})$?

2.For the training, IoU threshold is adaptively change. Why use this trick? It may lead to unfair comparison with other methods. Could you provide some experiments with fixed IoU threshold?

3.IoU loss is kept in the training. Why keep it? Could you provide some experiments without it to demonstrate its benefits?

Questions during rebuttal period:

Please address and clarify the cons above

Some typos:

(1)3.1: $\theta \in \begin{Bmatrix} 0, & \frac{\pi}{2}  \end{Bmatrix}  \to \theta \in (0, \frac{\pi}{2})$

(2)4.1 5th line: four types -> three types

---

> ### Author Response · Authors · 2020-11-25
> **Response to Reviewer 4**
>
> i) $\theta$ use
>
> Thanks for your comments!  For your first question, the value of $\theta$ is limited in $[0, \frac{\pi}{2}]$, and thus we use the tangent function to project its value into the $R$ space. Some numerical technologies can be used to avoid gradients exploding at the beginning of the training process.
>
>
>
> ii) IoU change during training
>
> We apologize that it's a typo and we will remove it in the revision.
>
>
>
> iii) why we need two loss functions
>
> We can directly predict objects by learned corners. However, it fails to differentiate the quality of the positive predictions (quality means the IoU with objects), while the GIoU loss can push predictions to high quality, which is important to the COCO datasets. If the GIoU loss is removed, the performance drops from 39.3 to 36.5, indicating the importance of the GIoU loss.

---

### Official Review · AnonReviewer3 · 2020-10-27

**Rating:** 3
**Confidence:** 5

**Review:**

##########################################################################

Summary:

This paper proposes an anchor-free object detector that does bounding box regression in the polar coordinate instead of in the Cartesian coordinate. The motivation of doing this is because there are larger variance in offset vectors in the Cartesian coordinate (the extreme case when a point is on one of the four corners of the bounding box, the offset vector becomes [0, w, 0, h]). The authors propose a solution to regress to the pair of corners (either TL+BR or TR+BL) in the polar coordinate, and select the corner pair that gives the smallest variance during training.

##########################################################################

Pros:

Experiment results show using polar coordinate is effective.

##########################################################################

Cons:

1. The authors did some analysis on the variance of the offset vector in Section 3.1, however, I think the analysis is not enough. First, the analysis only contains the worst case analysis, that is, the range of offset targets. And the author directly concluded from this: PolarNet "significantly reduces the variance" (page 4 last line). What is the ratio of variance under cartesian and polar coordinate to make it "significant"? I do not see any number either theoretically proves it or empirical analysis of the variance during training.

2. The term "keypoint" used in this paper is confusing. Sometimes the "keypoint" refers to corner points of the box ("Keypoint Position") and other times the "keypoint" simply refers to any point within the bounding box ("Keypoint Offsets").

3. The introduction of PolarNet is not clearly presented, specifically there are several confusing points:
- Section 3.4.1, what is the usage of t_{x,y}? I don't see how t is used during inference.
- Section 3.5, "we select the optimal box from b_{x,y} as the final output of the predicted box", what is "optimal box"?
- Figure 3, why the "corner supervision" comes from the feature map? I don't see how corner supervision uses any feature.
- The proposed corner supervision is simply the L1 loss, and there are methods that already use it with IoU loss. I don't think it is a contribution and section 3.4.2 and 3.4.3 should be combined.

4. Experiments are not solid:
- There are ways to reduce variance of offsets under cartesian coordinate, e.g. only use points within the center region of the bounding box to learn offset. Such experiment should be compared.
- The importance of extra loss function is also not studied, what are the benefit of using more losses?
- From reading Section 3, I feel the method is exactly as applying FCOS + polar coordinate, but Table 2 shows there is still some gap. Where does the extra gain come from?
- I checked the FCOS paper and found the R101 results in the paper is 43.2 but the number in this paper is 41.5.

##########################################################################

Reasons for score:

Overall, I vote for rejecting. This paper proposed an interesting idea, but I think way it is presented is not good enough to be accepted. Specifically, I think the paper still misses analysis on the variance of offset prediction, and also misses some important ablation studies. Furthermore, the paper is not well-written and requires some revision.

---

> ### Author Response · Authors · 2020-11-25
> **Response to Reviewer 3**
>
> i) Insufficient analysis or results of offsets variance during training process
>
> It’s non-trivial to learn which specific ratio is optimal to learn detectors, so we plot the heatmaps of ratio encoded by vanilla FCOS and PolarNet in Appendix. From the heatmaps we can see for FCOS, only keypoints within a small central region, the variance is small, and for PolarNet, the low variance region is much larger, indicating we have more high quality keypoints to use for training.
>
> ii) Keypoints mis-usage
>
> The termination “keypoints” in the paper denotes each point used to present objects. The selection of the point depends on which detection algorithms you use. For example, in CornerNet, corner points are selected as keypoints, and in FCOS, all points within the objects are selected as keypoints.
>
> iii) PolarNet is not clearly represented.
>
> During PolarNet training, each point has two pairs of corners, and we select the corner pair with similar distances to the point as the optimal output. The motivation is to discard the poor quality corner pair during training. t_{x,y} is used to predict which corner pair to use during inference. And the “optimal box” in your question denotes the box generated by the selected corner pair. We will add more details in the revision.
>
> iv) No features from corners are used and L1 supervision has already been used with GIoU loss before which is no contribution
>
> We apologize for the lack of some details in corner supervision, and we want to clarify a possible confusion regarding the corner supervision. During the training of PolarNet, corner features are used to enhance the original feature representation. Since PolarNet needs to learn the distance of keypoints towards their corresponding corners (Formula 6,7), thus we aim to use the scale-invariant corner features to enhance the original feature map. During training, we first learn the corner features and then we integrate them with the original feature map to make it more robust to scales.  Finally we learn classifiers and regressors based on this new feature map.
>
> So the figure is correct and it’s not a naive L1 loss. We will add more details in the revision.
>
> v) Experiments is not solid.
>
> a. The comparison with FCOS-CenterSampling
>
> We train FCOS models with points in the central region, and we show the results are 38.1 on COCO. Central region sampling can relieve the variance issue to some extent, but it is still worse than our method (39.3). In addition, we argue the region area of the central region is  manually designed which may impact the detection performance.
>
> b. Why use multiple losses
>
> We can directly predict objects by the learned corners. However, it fails to differentiate the quality of the positive predictions (quality means the IoU with objects), while GIoU loss can push predictions to high quality, which is important to the COCO datasets. If the GIoU loss is removed, the performance drops from 39.3 to 36.5, indicating the importance of GIoU loss.
>
> c. Where is the gain from FCOS+Polar to PolarNet
>
> The gain from “FCOS+Polar” to “PolarNet'' comes from the corner features. We do not integrate corner features in “FCOS+Polar”.
>
> d. Baseline of FCOS is 43.2
>
> FCOS reports 43.2 in X-101, but we use R-101 in Table 1, which reports 41.5 in original paper.

---

### Official Review · AnonReviewer2 · 2020-10-28
**A new and interesting keypoint-based object detector**

**Rating:** 8
**Confidence:** 5

**Review:**

This paper proposes a new key-point based object detector, PolarNet, which predicts the distances between key-points and corner pairs (such as top-left and bottom-right pair or top-right and bottom-left pair) on polar coordinates. This is different from other key-point based object detectors such as FCOS which predicts distances between key-points and bounding box boundaries on cartesian coordinates.  The authors claim that the advantage of representing the offsets in the form of polar coordinates is this representation reduces the variance in the offsets, which makes learning easier.

Pros:
This is an interesting approach and new, to the best of my knowledge, in the context of object detection. The authors show that PolarNet outperforms other approaches such as FCOS and FoveaBox under the same backbone network, ResNet-101. The use of polar coordinates improves the performance of FCOS by more than 4% which shows the effectiveness of the polar coordinates. With a larger backbone and deformable convolution, PolarNet demonstrates state-of-the-art performance among all anchor-free detectors on the challenging COCO dataset.

Cons:
I am confused about the corner supervision in section 3.4.2. It seems to me that the corner supervision is to train PolarNet to predict the offsets which are in polar coordinates. But the authors also apply this to FCOS (FCOS + Centerness + Corner) and compare it with FCOS with polar coordinates (FCOS + Polar).  I don’t understand the difference between them. How do the authors apply corner supervision if FCOS is predicting on cartesian coordinates (i.e. FCOS + Centerness + Corner)? How is that different from FCOS + Polar exactly? Or do I misunderstand the meaning of corner supervision? The authors also mention in the section where they introduce corner supervision that they train a regressor based on corner features. What does corner supervision mean exactly? Does it mean the authors extract features from the corners and use the features together with the key-point features when they predict the offsets? Or do they simply refer to the regression loss function?

The use of IoU loss seems to a bit redundant. It seems that the $L_{corner}$ already trains the network to predict the offsets. Why do the authors still need the IoU loss? The authors should provide an ablation study to demonstrate how the IoU loss is affecting the performance of PolarNet.

---

> ### Author Response · Authors · 2020-11-25
> **Response to Reviewer 2**
>
> i) “FCOS+Centerness+Corner” is the same as “FCOS+Polar”  and Details of corner supervision
>
> Thanks for your comments! First of all, We want to clarify a possible confusion regarding the corner supervision. PolarNet needs to learn the distance from the keypoints to their corresponding corners (Formula 6,7), and thus the corner representation is important. Here we aim to use the scale-invariant corner features to enhance the original feature map, and we first learn the corner features and then we integrate these corner features with the original feature map to make it more robust to scales.  Finally we learn classifiers and regressors based on this new feature map.
>
> “FCOS+Centeness+Corner” means we train a FCOS model with the integrated corner features, while “FCOS+Polar” means we train a FCOS with Polar coordinates but we do not use corner features. And finally PolarNet uses both corner features with polar coordinates to predict objects.
>
> ii) Reason to use two loss functions
>
> Thank you for your comment,  we can directly predict objects by learned corners. However, it fails to differentiate the quality of the positive predictions (quality means the IoU with objects), while GIoU loss can push predictions to high quality, which is important to COCO datasets. If GIoU loss is removed, the performance drops from 39.3 to 36.5, indicating the importance of GIoU loss.

---

### Official Review · AnonReviewer1 · 2020-11-03
**Intuitive design with reasonable performance gain**

**Rating:** 6
**Confidence:** 4

**Review:**

# Summary
In this paper, the authors propose a simple but effective keypoint-based anchor-free object detection system. The main idea is to replace the Cartesian coordinate with the polar coordinate, compared to the closest related work, FCOS. According to the extensive empirical results, the proposed system achieves a better trade-off between speed and accuracy.

The overall writing looks good to me. The storyline is consistent and well-motivated. The authors provide enough detail to shed light on the design choices for the state-of-the-art anchor-free detector. The figures and tables are also quite informative. For example, I particularly love the figure 1 because it helps the readers catch up with the most recent progress on keypoint-based object detection frontier. It could be better if the authors could describe more details in the caption. By the way, the black color of rho and theta should be changed into a lighter one.

# Questions

1. scale-sensitive vs. scale-invariant
In this paper, the author mentioned the scale-related terminology many times. I wonder if the authors could explain more detail about why PolarNet is scale-invariant? From my perspective, the offset regression is scale-sensitive because the target numbers are strongly related to the actual object size. Even though the proposed method utilizes the corner points to localize objects, the bounding box offset part is still scale-sensitive, right?

2. center-based method + polar coordinate
I wonder if the authors could try to put the polar coordinate offset regression into a center-based anchor-free detector. For example,  CenterNet[1] regresses the bounding box offset in the Cartesian coordinate system. What if we change the distance encoding to polar coordinate? In section 4.4, the authors claim, "Speciﬁcally, compared with other center-based methods such as CenterNet or FSFA, our method not only extracts features from the central region, but also encodes features from the whole bounding boxes." I wonder how much performance gain could be seen if we only change the coordinate system. It could be a helpful ablation study to support this claim.

# Reference
[1] Zhou, Xingyi, Dequan Wang, and Philipp Krähenbühl. "Objects as points." arXiv preprint arXiv:1904.07850 (2019).

---- Post-rebuttal comments----

The rebuttal and the paper revision address my concerns. I keep my original rating.

---

> ### Author Response · Authors · 2020-11-25
> **Response to Reviewer 1**
>
> i) PolarNet is still scale-variant(sensitive)
>
> Thanks for the comments! Here we emphasize the scale-invariant representation denotes the corner feature, and the PolarNet framework is not scale-invariant but scale-insensitive. The corner feature is scale-invariant and is used to make PolarNet robust to scale variance than existing anchor-free methods. In addition, the keypoints’ offsets are encoded with polar coordinates which significantly reduce the variance to avoid poor quality cases in Cartesian variants, making PolarNet scale-insensitive.
>
> ii) CenterNet + Polar?
>
> Thank you for your suggestion, we re-train PolarNet with only center points sampled for training. The result in COCO based on R-101 is 39.7, which is better than CenterNet (reported 34.6 in their paper), but much worse than our full version (43.6). This result indicates the polar system is effective and can bring more positive keypoints for training.

---

### Decision · Program_Chairs · 2021-01-07
**Final Decision**

**Decision:**

Accept (Poster)

**Comment:**

This paper received overall positive scores. One reviewer (R3) recommended clear reject.

All reviewers agree that the paper introduces a novel idea and its effectiveness is supported by the experimental results. There are concerns about clarity of presentation and certain missing analyses, which have been addressed by the authors in the rebuttal. Thus the ACs recommend acceptance.